# The Impact of Long Noncoding RNAs in Tissue Regeneration and Senescence

**DOI:** 10.3390/cells13020119

**Published:** 2024-01-09

**Authors:** Júlia Tavares e Silva, João Pessoa, Sandrina Nóbrega-Pereira, Bruno Bernardes de Jesus

**Affiliations:** Department of Medical Sciences and Institute of Biomedicine—iBiMED, University of Aveiro, 3810-193 Aveiro, Portugal; juliamarisasilva@gmail.com (J.T.e.S.); joao.pessoa@ua.pt (J.P.); sandrina.pereira@ua.pt (S.N.-P.)

**Keywords:** aging, cellular senescence, long noncoding RNAs, upregulation, downregulation, neurodegeneration, cardiovascular disease

## Abstract

Overcoming senescence with tissue engineering has a promising impact on multiple diseases. Here, we provide an overview of recent studies in which cellular senescence was inhibited through the up/downregulation of specific lncRNAs. This approach prevented senescence in the bones, joints, nervous system, heart, and blood vessels, with a potential impact on regeneration and the prevention of osteoarthritis and osteoporosis, as well as neurodegenerative and cardiovascular diseases. Senescence of the skin and liver could also be prevented through the regulation of cellular levels of specific lncRNAs, resulting in the rejuvenation of cells from these organs and their potential protection from disease. From these exciting achievements, which support tissue regeneration and are not restricted to stem cells, we propose lncRNA regulation through RNA or gene therapies as a prospective preventive and therapeutic approach against aging and multiple aging-related diseases.

## 1. Introduction

The aging of the human population is correlated with the increased prevalence of multiple diseases, including neurodegenerative [1] and cardiovascular [2] disease, cancer [3], and others, which represent large healthcare and economic burdens. Aging can be defined as a progressive decline of the physiological functions required for survival and fertility, due to the buildup of cellular damage [4]. The hallmarks of aging comprise stem cell exhaustion, altered intercellular communication, genomic instability, telomere attrition, epigenetic alterations, loss of proteostasis, deregulated nutrient sensing, mitochondrial dysfunction, and cellular senescence [4]. The possibility of suppressing cellular aging and promoting tissue regeneration through targeting senescent cells makes this aging hallmark a promising and exciting subject of research.

Cellular senescence is characterized by an irreversible cell cycle halt that can be caused by critical telomere shortening and multiple damaging stimuli, including DNA damage, oncogene activation, oxidative stress, and chemotherapy [5,6]. Senescence contributes to tissue remodeling via the senescence-associated secretory phenotype (SASP), a combination of factors released by senescent cells and including several families of soluble and insoluble factors [7,8]. Despite the role of senescent cells in tissue homeostasis, their accumulation with aging results in age-related diseases and tissue loss of function [9], including alterations in the replicative potential of stem cells [9].

Given the association between cellular senescence and aging, tissue engineering approaches that target senescent cells are promising strategies to promote regeneration, delay aging and limit age-related diseases [10]. The two main anti-senescence approaches are senolytics, a class of drugs that selectively induce senescent cell apoptosis, and senomorphics, drugs that interfere with SASP induction, maintenance, and activity [11]. Despite their promising therapeutic potential, senolytics exhibit reduced efficacy and significant toxicity [12]. Furthermore, senomorphics require regular and prolonged administration to yield benefits [13]. Alternative strategies would be beneficial for the prevention and treatment of multiple highly prevalent human diseases related to aging. Here, long noncoding RNAs (lncRNAs) were shown to be deeply involved in cellular senescence programs [14,15,16], being promising molecules for its precise regulation.

LncRNAs are a class of transcripts that regulate gene expression at multiple levels, being involved in multiple cellular processes, including the cell cycle, cell differentiation, and apoptosis [17]. In addition to their roles in multiple diseases, including cancer and metabolic, nervous system, and cardiovascular disorders [18], lncRNAs are also involved in cellular senescence [19]. They mediate the nuclear architecture alterations of senescent cells [20], affect SASP composition [21], and can induce or repress senescence [14]. Aging-associated lncRNAs seem to be more conserved than age-independent lncRNAs [22] and can be identified through RNA sequencing analysis [23]. Centenarians express increased levels of specific lncRNAs that decrease the levels of senescence markers including p16, p21, and senescence-associated β-galactosidase activity [24]. Recent reviews have detailed the impact of multiple lncRNAs on cellular senescence and aging-related diseases [14,15,25,26,27,28,29]; however, their detailed molecular mechanisms are yet to be uncovered [15].

One of the most promising resources in tissue engineering approaches to treat pathologies, including age-related diseases, are stem cells [30]. These cells are well-known for their self-renewal and differentiation abilities [31], which enable the development and regeneration of organs and tissues [32]. Stem cells, including embryonic stem cells (derived from the inner cell mass of pre-implantation embryos) [33] and induced pluripotent stem cells (obtained from somatic cells through cell reprogramming) [34], can be permanently maintained in their pluripotent state in vitro, while preserving their ability to differentiate under appropriate signal inputs [35]. However, and although lncRNAs have been shown to impact the reprograming process [16,36], their use has important limitations. These limitations include the complexity and time requirements of cell reprogramming and cell heterogeneity among pluripotent stem cell lines. Moreover, the risks of tumor formation and immune rejection after transplantation should not be underestimated [37].

In the present review, we compile recent evidence on the role of lncRNAs in cell senescence with an impact on tissue regeneration. Tissue engineering involving lncRNA regulation is promising for the prevention and treatment of multiple diseases, being a prospective approach for the control of a vast array of human diseases.

## 2. General Features of lncRNAs

Non-coding RNAs (ncRNAs) comprise around 98.5% of the transcriptome of multicellular eukaryotes [38]. Although not encoding proteins, ncRNAs present high tissue specificity, playing a key role in their function and identity. These transcripts also act as transcriptional and post-transcriptional regulators of the coding transcriptome. Therefore, the regulation of these RNAs is of high importance as they can contribute to the development of several pathologies [16,39].

The non-coding transcriptome includes a diverse range of RNA species that can be classified into two major classes based on their size: small ncRNAs and lncRNAs. Small ncRNAs, which include microRNAs, small interfering RNAs, small nucleolar RNAs, and piwi-interacting RNAs, play diverse biological roles. These transcripts are involved in guiding chromatin modifications and, through interaction with other molecules, can regulate gene expression through RNA interference, RNA modification, or spliceosomal involvement [40,41]. LncRNAs, on the other hand, comprise the most extensive class of non-coding transcripts, and cover RNA species with a length greater than 200 nucleotides, but can frequently extend to several kilobases [40].

LncRNAs comprise RNAs that are transcribed by RNA polymerase I (Pol I), Pol II, and Pol III and are distributed across the entire genome, classified as intergenic (long intergenic non-coding RNAs), antisense (natural antisense transcripts), intronic, or pseudogenes. These non-coding transcripts can also arise from back-splicing of coding and non-coding transcripts from *trans*-acting regulatory RNAs and can change their structure to increase their stability [16,42]. The majority of lncRNAs are transcribed, spliced, and exported to the cytoplasm; however, many Pol II-transcribed lncRNAs are inefficiently processed and retained in the nucleus. Although highly involved in the regulation of gene expression at different levels, their specific functions depend on their localization as well as on their interactions with proteins, DNA, and RNA. Among their many functions, they can modulate chromatin structure and function, modulate the transcription of nearby and distant genes, and intervene in signaling pathways. They can also have an impact on RNA splicing, stability, and translation and participate in the regulation and establishment of organelles and nuclear organization [17]. Despite their distinct properties, lncRNAs and protein-coding genes share similar regulatory mechanisms of expression. They both undergo DNA methylation and histone modification, sharing the same transcription factors and chromatin modifications. Their characteristic low expression levels and the cell and tissue type specificity observed in lncRNAs are also directly linked to DNA methylation and histone modification [43].

Due to their tissue specificity and their crucial role in epigenetic modulation, the dysregulation of lncRNAs is usually associated with the onset of diseases. Therefore, lncRNAs may constitute a potential therapeutic target for numerous pathologies. However, most lncRNAs do not have conserved sequence motifs that predict their biological functionality, which poses a challenge for their study in an in vivo context. Furthermore, lncRNA sequences are typically not conserved across different species, making it hard to study these transcripts [44,45]. This lack of conservation represents a major obstacle for the study of lncRNAs as therapeutic tools, as a lncRNA with therapeutic potential in mice may not have the same potential in humans.

## 3. LncRNAs in Tissue Engineering

The applications of lncRNAs in tissue engineering may include a myriad of strategies [46,47]. Here, we describe the impact of lncRNAs in cell identity in different biological systems, in particular the role of lncRNAs in the pathways governing age-dependent tissue degeneration or regeneration.

### 3.1. Neurodegeneration

Brain aging is correlated with an increased prevalence of neurodegenerative diseases, which are aggravated by the senescence of nervous cells. Interestingly, multiple preclinical studies have demonstrated that increasing or decreasing the levels of specific lncRNAs in cells of the nervous system could counteract their senescence (Table 1), which could be a valuable approach to delaying neurodegeneration.

A combination of cognitive and physical exercises could delay cognitive decline in naturally aging rats. This effect was mediated by a decrease in the hippocampal levels of the nuclear-enriched abundant transcript 1 (*Neat1*) lncRNA [48]. This study demonstrates the possibility of lncRNA downregulation through physical and cognitive activities in rats. These activities also had a positive impact on their hippocampal function. In aged mice, the levels of growth-arrest-specific transcript 5 (*Gas5*) were increased in the hippocampus, suggesting that it could drive cellular senescence. Nevertheless, its upregulation decreased senescence markers [49], indicating that *Gas5* could actually protect against senescence in the hippocampus. Another study demonstrated that human primary astrocytes became senescent upon exposure to the HIV transactivator of transcription. Such increased senescence levels were correlated with the upregulation of taurine-upregulated gene 1 (*TUG1*) lncRNA. Interestingly, *TUG1* downregulation decreased the levels of senescence markers [50]. These studies demonstrate that the levels of specific lncRNAs in the central nervous system change with aging and could be up- or downregulated to delay the senescence of cells from the nervous system.

LncRNAs have been proposed as potential biomarkers for the diagnosis of Parkinson’s disease [55]. The overexpression and aggregation of α-synuclein are hallmarks of this disease [56]. Of note, α-synuclein mRNA and protein levels were upregulated by the transcription of synuclein alpha antisense 1 (*SNCA-AS1*), an antisense lncRNA of the α-synuclein-coding gene. *SNCA-AS1* upregulation in cultured cells decreased the RNA levels of synapse markers and increased those of most senescence markers tested. *SNCA-AS1* knockdown had opposite effects [53]. With α-synuclein being a key player in Parkinson’s disease, this study puts forward the downregulation of its antisense lncRNA as a potential therapeutic strategy.

These studies demonstrate that multiple lncRNAs can be up- or downregulated in cells from the nervous system, with positive outcomes in the prevention of their senescence and a potential impact on promoting regeneration and limiting neurodegeneration.

### 3.2. Osteoarthritis and Osteoporosis

Additional health issues related to aging are the loss of cartilage and bone mass, which cause osteoarthritis and osteoporosis, respectively. While osteoarthritis is a painful condition caused by decreased joint mobility, osteoporosis increases bone fragility and the risk of fracture. Senescence of cartilage or bone cells is involved in these diseases [57]. Interestingly, lncRNA up/downregulation is also a promising strategy against osteoarthritis and osteoporosis, promoting bone regeneration (Table 2).

Increased levels of epidermal growth factor receptor long noncoding downstream RNA (*ELDR*) were detected in the chondrocytes (joint cartilage cells) of osteoarthritis tissues, where they could drive chondrocyte senescence. Downregulation of the *Eldr*-coding gene in mice promoted chondrocyte proliferation, inhibiting senescence and cartilage degeneration. Its inhibition in explants from osteoarthritis patients could also suppress senescence markers [58]. Osteoarthritis is also related to decreased chondrocyte levels of the zinc finger homeobox 2 (*lncZFHX2*) lncRNA, whose depletion accelerated the senescence of these cells. *LncZFHX2* was upregulated under physiological hypoxia in chondrocytes from both human and mouse cartilage; however, in mice, *lncZfhx2* levels naturally decrease with age, increasing the risk of osteoarthritis [59]. The Fer-1-like protein 4 (*FER1L4*) lncRNA was also downregulated in osteoarthritis patients. Its levels also decreased with aging and disease progression [60]. The *AC006064.4-201* lncRNA alleviated chondrocyte senescence and protected against osteoarthritis, being downregulated in senescent human cartilage. Its downregulation results in increased stability of the mRNA coding for cyclin-dependent kinase inhibitor 1B, which has a pro-senescence impact [61]. Similarly to osteoarthritis, osteoporosis is related to excessive cellular senescence. In osteoporosis, the senescence of osteoblast progenitor cells impedes their differentiation into osteoblasts and migration to the bone formation sites, resulting in the loss of bone mass. Specific lncRNAs have been demonstrated to be promising therapeutic targets to combat osteoporosis through their therapeutic regulation. Knockdown of the zinc finger antisense 1 (*Zfas1*) lncRNA suppressed cell senescence and promoted osteogenic differentiation, which increased bone mass [62]. In bone marrow mesenchymal stem cells from aged mice, postulated migration inhibitory factor (*Pmif*) lncRNA was highly expressed and caused a reduction in the migration capacity of osteoprogenitor cells to the bone formation site. *Pmif* knockdown promoted bone formation in aged mice [63]. In aged bone marrow mesenchymal stem cells, *NEAT1* was upregulated, promoting osteoclast differentiation and impairing osteogenesis. Consistently, *NEAT1* knockdown prevented age-related bone loss [64]. These studies show that lncRNAs mediate osteoblast differentiation and migration to the bone formation surfaces, as well as osteoclast differentiation, indicating these lncRNAs as potential therapeutic targets against osteoporosis. While lncRNAs regulate gene expression that affects cartilage health, they also mediate osteoblast differentiation and migration, which are critical for the generation and regeneration of bone mass. In both systems, regulation of the levels of specific lncRNAs could prevent senescence that would otherwise contribute to osteoarthritis or osteoporosis.

### 3.3. Cardiovascular Diseases

The lncRNA transcriptome changes with heart aging, these alterations being correlated with an increased prevalence of cardiovascular diseases. The human endothelial transcriptome has about 17% of its lncRNAs significantly altered in advanced age [65]. Multiple lncRNAs are related to the aging of endothelial cells and cardiovascular diseases [66,67,68]. LncRNAs are potential therapeutic targets against endothelial senescence, with a potential impact on vascular aging [69]. As described below, multiple recent studies have demonstrated the possibility of overcoming senescence in cells from the heart and blood vessels through alterations in the levels of senescence-associated lncRNAs (Table 3).

The levels of heart-enriched senescence-mitophagy-associated lncRNA (*lncRNA-SMAL*) were increased in humans over 60 years old. *LncRNA-SMAL* promoted senescence in cardiomyocytes, and its inhibition had a cardioprotective effect [70]. The *H19* lncRNA usually functions as a microRNA sponge to control gene expression, being implicated in multiple diseases related to aging and inflammation [81]. *H19* levels increased in senescent cardiomyocytes and in aged mouse hearts. Its downregulation protected against cardiomyocyte senescence [71]. The levels of lncRNA cytochrome P450 family 7 subfamily A member 1 (*lnc-CYP7A1-1*) also increased with aging in human bone-marrow-derived mesenchymal stem cells. Downregulation of *lnc-CYP7A1-1* in these cells inhibited their senescence and improved cardiac function after implantation into the hearts of mice recovering from myocardial infarction [72].

Overcoming senescence through the regulation of lncRNAs has also been demonstrated in endothelial cells, with promising outcomes. The 17β-estradiol hormone could inhibit H_2_O_2_-induced senescence in human umbilical vein endothelial cells (HUVECs). It functioned by increasing the levels of plasmacytoma variant translocation 1 (*PVT1*), an anti-senescence lncRNA that could be upregulated through 17β-estradiol [73]. Decreased levels of *ENSMUST00000218874* lncRNA were found in the aortas and vascular smooth muscle cells of aged mice, which could show an anti-senescence role for this lncRNA, which was confirmed through its knockdown in cells [74]. Prostate-cancer-associated transcript 14 (*PCAT14*) was found to be relatively abundant in young human endothelial cells, but downregulated with age. *PCAT14* promoted endothelial regeneration and angiogenic capacity, having potential anti-senescence properties [65]. The levels of the *GAS5* lncRNA were decreased in the serum of aged patients and in the central arteries of aged rats. *GAS5* overexpression in vascular smooth muscle cells decreased their senescence levels [75]. Low levels of the small nucleolar RNA host gene 1 (*SNHG1*) lncRNA in vascular smooth muscle cells were related to their senescence. Overexpression of *SNHG1* decreased the calcification and senescence of these cells [76]. Deep venous thrombosis is related to senescence of the endothelium and decreased expression of silent information regulator 1 (*SIRT1*) mRNA and its antisense lncRNA, *SIRT1-AS*. Overexpression of *SIRT1-AS* increased *SIRT1* levels and decreased senescence and deep venous thrombosis markers in human vascular endothelial cells [77]. These studies identify lncRNAs whose decreased levels in blood vessel cells were correlated with senescence and aging. Their overexpression could improve cardiovascular function.

In contrast, excessive levels of other lncRNAs have been related to endothelial cell senescence. The antisense transcript 1 of receptor-activity-modifying protein 2 (*RAMP2-AS1*) lncRNA promoted the expression of a protein-coding gene, *RAMP2*. However, *RAMP2-AS1* knockdown induced senescence in endothelial cells [78]. In HUVECs, the levels of maternally expressed gene 8 (*MEG8*) lncRNA increased with passage number, suggesting its upregulation with aging. Its knockdown slightly increased senescence and impaired the endothelial barrier, indicating a protective role for this lncRNA in vascular aging [79]. Opa-interacting protein 5 antisense RNA 1 (*Oip5-AS1*) levels were increased in the aortas of aged mice. Its downregulation in HUVECs decreased their H_2_O_2_-induced senescence [80]. In contrast with the data from the previous paragraph, some lncRNAs induce the senescence of blood vessel cells when their levels are increased. Their downregulation has revealed positive outcomes.

### 3.4. Skin Aging

LncRNAs are also involved in skin aging, being potential therapeutic targets to revert this process [82,83]. In fact, lncRNA regulation has been successfully tested to overcome the aging of skin cells and promote their regeneration (Table 4).

Specifically, *H19* lncRNA levels were decreased in aged human dermal fibroblasts. Their upregulation in these cells had a protective effect against senescence and enhanced cell viability [84]. On the other hand, the downregulation of the membrane-associated guanylate kinase WW and PDZ domain-containing 2 antisense RNA 3 (*MAGI2-AS3*) in human fibroblasts had a delaying effect on senescence [85]. LncRNAs could also be useful in wound healing. The lncRNA senescence-associated noncoding RNA (*SAN*) was overexpressed in aged adipose-derived stem cells. Its knockdown inhibited the senescence of these cells and enhanced their function. Furthermore, the transplantation of *SAN*-depleted adipose-derived stem cells accelerated wound closure in rats [86]. On the other hand, the lysophospholipase-like 1 antisense RNA 1 (*LYPLAL1-AS1*) lncRNA was downregulated in senescent human adipose-derived mesenchymal stem cells and in human blood from middle-aged individuals (relative to younger ones). *LYPLAL1-AS1* overexpression could revert senescence in human adipose-derived mesenchymal stem cells [87]. However, this study did not test the impact of *LYPLAL1-AS1* overexpression on wound closure. Nevertheless, the above-mentioned research improves our understanding of the mechanisms of skin aging and its regeneration. Specific lncRNAs could be up- or downregulated in fibroblasts to rejuvenate these cells. Furthermore, lncRNA regulation in adipose-derived mesenchymal stem cells followed by their transplantation might improve rates of wound closure; however, more studies are needed to robustly assess this hypothesis.

### 3.5. Liver Diseases

Aging of the liver is related to increased incidence and faster progression of hepatic diseases including nonalcoholic fatty liver disease (NAFLD) and hepatocellular carcinoma [88]. Specific lncRNAs have been shown to affect liver aging and NAFLD pathophysiology (Table 5).

T_reg_-expressed non-protein-coding RNA (*Altre*), produced by regulatory T cells (T_reg_), was upregulated with aging and protected the liver against aging-related diseases. Its deletion in T_reg_ cells had a severe impact on the livers of aged mice, but not in younger animals [89]. These findings suggest *Altre* is a potential protective molecule against liver aging and related diseases. While senescence and aging are deeply related, senescence can impact diseases that are not directly related to the aged population, such as NAFLD. NAFLD is an extremely prevalent disease, which, in its initial stage, consists of hepatic steatosis (fat accumulation) but can subsequently evolve to more severe stages manifesting mitochondrial dysfunction, oxidative stress, and other deleterious effects on liver cells [92]. Of note, specific lncRNAs have shown protective roles against the negative impact of senescence on NAFLD liver cells. The maternally expressed gene 3 (*Meg3*) lncRNA was increased in the hepatic endothelium of obese mice and in the livers of human patients with NAFLD. *Meg3* knockdown in diet-induced obese mice caused senescence of the hepatic endothelium, promoted insulin resistance, and impaired glucose metabolism. These effects could be overcome through the knockout of p53 in the hepatic endothelium [90]. Hepatocyte senescence has also been detected in cellular and animal models of NAFLD. The MIST-1/2 antagonizing for YAP activation (*MAYA*) lncRNA promoted NAFLD through iron overload, fat accumulation, and senescence in hepatocytes. Its downregulation in a hepatocyte cell line alleviated these effects [91]. These studies confirm that NAFLD, which is mostly caused by poor lifestyle habits, is also negatively impacted by liver cell senescence. LncRNAs are potential tools to decrease liver cell senescence, regenerate liver function, and protect the liver from diseases that are aggravated by cellular senescence.

## 4. LncRNA Transduction

Cellular senescence is implicated in conditions including neurodegeneration, osteoarthritis, osteoporosis, cardiovascular diseases, liver diseases, and skin aging (Table 1, Table 2, Table 3, Table 4 and Table 5). Our literature searches globally indicate that, in these conditions, the equilibrium between cellular senescence/aging and tissue regeneration can be controlled by regulating the levels of specific lncRNAs. Therefore, senescence-associated lncRNAs are potential targets for therapeutic up/downregulation in multiple aging-related human diseases. Globally, their levels can be modified to overcome multiple disease features and improve the functioning of the organism (Figure 1).

The upregulation of lncRNAs has usually been achieved through their overexpression from an exogenous gene, while their downregulation has frequently employed RNA oligonucleotides, such as small interfering RNA (siRNA), antisense oligonucleotides, short hairpin RNA (shRNA), or other molecules. There are critical factors that need to be overcome for clinical translation. First, lncRNA up/downregulation needs to be achieved only in the intended target cells, to avoid side effects. To achieve lncRNA regulation only in a specific cell type, ligands of receptors overexpressed on the surface of the target cells could be exploited. As exemplified in another research field, possible approaches could be fusing the drug to the ligand [93] or packing it into nanoparticles coated with ligands that bind the overexpressed cell surface receptor [94]. Delivery systems could be exogenous vectors, including viral vectors, liposomes, or lipid nanoparticles, or endogenous vectors, such as exosomes [95]. Second, efficient cellular uptake is also essential. In brain cells, this step is particularly challenging, due to the reduced permeability of the blood–brain barrier, which is an obstacle to the treatment of neurodegenerative diseases. Nevertheless, the blood–brain barrier contains transporters for peptides and proteins, which could be exploited for drug delivery strategies. The drug delivery vector could be modified with monoclonal antibodies mimicking substrates of these transporters, which might facilitate their crossing of the blood–brain barrier and delivery into the central nervous system [96]. Third, cellular uptake of exogenous compounds frequently occurs through endocytosis. The therapeutic compound needs to escape the endosomes and reach the required subcellular location. If endosomal escape is inefficient, the drug may be degraded inside the lysosomes, limiting its therapeutic potential. While regular transfection usually overcomes this limitation, the use of tailored vectors for delivery into a specific cell type may require its re-assessment. Fourth, RNA is intrinsically unstable. Furthermore, the hydrophilic nature and negative charge of oligonucleotides are obstacles to their membrane diffusion. To overcome these limitations, chemical modifications are effective in increasing RNA stability and membrane permeability. As such, all licensed RNA-based therapies utilize chemically modified RNA [97]. Fifth, any potential immunogenicity of the drug or its delivery system should also be tested. An immune response would cause degradation of the drug and a lack of therapeutic effect, with potential inflammatory responses. For example, single-stranded RNA molecules are usually more immunogenic than double-stranded RNA. Consequently, all siRNA-based therapies use double-stranded RNA [97]. Furthermore, the repeated administration of a lncRNA regulation therapy could also trigger an immune response, which would limit its use against chronic diseases. Nevertheless, the use of immunosuppressants might be considered; however, these drugs should be avoided in immunocompromised patients. A tissue engineering approach for lncRNA upregulation may include packing lncRNA-expressing genes into nanoparticles coated with ligands recognized by target cells (Figure 2A). For lncRNA downregulation, these particles would alternatively be loaded with suitable downregulatory oligonucleotides (such as siRNA, shRNA, or others) carrying chemical modifications to improve their stability (Figure 2B). The RNA instability limitation could be addressed through gene therapies using artificial genes coding for RNA oligonucleotides triggering lncRNA knockdown. This approach would benefit from the success of already approved gene therapies, such as those for melanoma, an aggressive type of skin cancer [98], or Leber congenital amaurosis, a genetic disease that causes blindness [99]. Overall, there are many upcoming challenges in the clinical translation of tissue engineering approaches based on lncRNA up/downregulation. Nevertheless, the high number of lncRNAs that have been successfully tested at the preclinical level demonstrates an equally high number of opportunities for successful therapy development.

## 5. Conclusions and Discussion

In the present review, we highlight that cellular senescence can be inhibited in multiple disease models by altering the cellular levels of specific lncRNAs. Senescence inhibition can also be correlated with the modulation of additional aging hallmarks and promote tissue regeneration. This approach may be beneficial for the prevention and treatment of multiple diseases triggered or aggravated by cellular senescence. Overall, LncRNAs can function as molecular switches for senescence in multiple cellular and disease models. These recent achievements open up exciting perspectives on the prevention and treatment of age-related traits during aging and disease.

## Figures and Tables

**Figure 1 cells-13-00119-f001:**
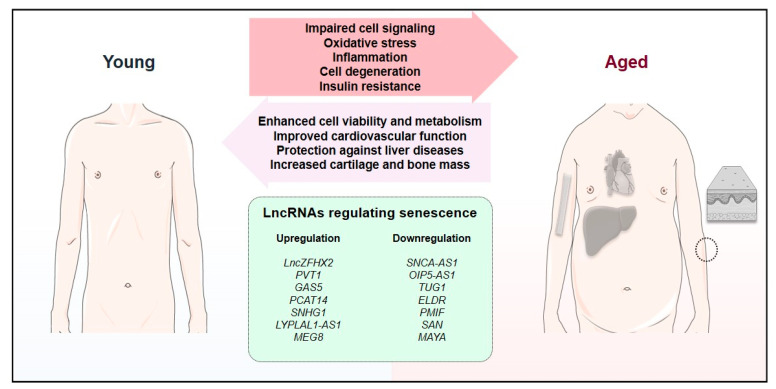
The impact of senescence-associated lncRNAs on the equilibrium between young and aged phenotypes. Aging is characterized by deleterious effects, including cell signaling dysfunctions, oxidative stress, inflammation, cell degeneration, and insulin resistance. Importantly, multiple lncRNAs can be regulated to revert these processes, resulting in enhanced cell viability and metabolism, as well as more specific effects, including improved cardiovascular function, protection against liver diseases, and increased cartilage and bone mass. In this figure, we included several examples of lncRNAs whose up- or downregulation could promote these effects though the inhibition of senescence.

**Figure 2 cells-13-00119-f002:**
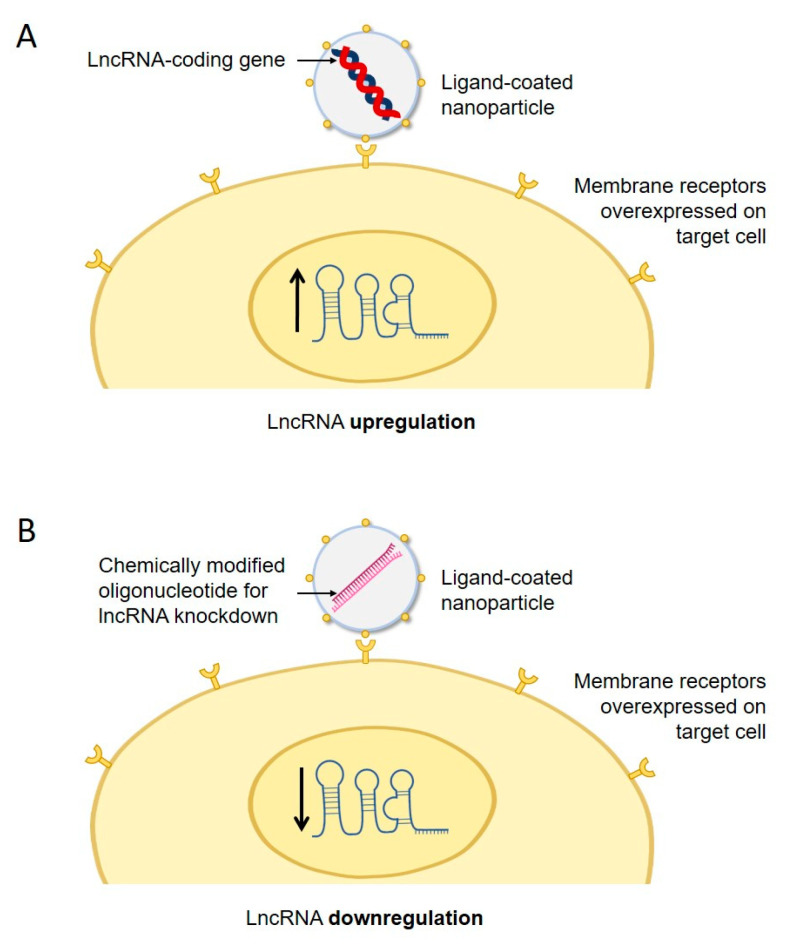
Possible tissue engineering strategies for therapeutic up/downregulation of lncRNAs. Therapies could be based on nanoparticles coated with ligands recognizing a surface receptor overexpressed on target cells. (**A**) For upregulation, nanoparticles could be loaded with a lncRNA-expressing gene. (**B**) For downregulation, nanoparticles could be loaded with chemically modified lncRNA downregulatory oligonucleotides. LncRNA upregulation or downregulation are represented by upward or downward arrows, respectively.

**Table 1 cells-13-00119-t001:** Studies on the impact of lncRNA up/downregulation on senescence of cells from the nervous system.

LncRNA	Associated Disease	Model Systems	Senescence Markers Assessed	Interacting Molecules and Molecular Mechanism	Up/Downregulation Outcomes	Reference
*Neat1*	Not specified	Naturally aging Sprague–Dawley rats, HT22 mouse hippocampal cells	None	*Neat1* downregulates the miR-124–3p microRNA; miR-124–3p upregulates the caveolin-1-PI3K/Akt/GSK3β signaling pathway	*Neat1* downregulation in rats during physical and cognitive exercises and in HT22 cells inhibited the caveolin-1-PI3K/Akt/GSK3β signaling pathway	[48]
*Gas5*	Not specified	HT22 hippocampal neuronal cells, male C57BL/6 mice	P16, p21	In mice, hippocampal *Gas5* levels naturally increase with age	Levels of p16 and p21 were slightly increased or decreased in HT22 cells under *Gas5* up- or downregulation, respectively	[49]
*TUG1*	HIV infection	Human primary astrocytes, *Hiv-1* transgenic rats	P16, p21, senescence-associated β-galactosidase	*Hiv-1* transgenic rats and human primary astrocytes exposed to the HIV transactivator of transcription have increased levels of *TUG1* and senescence markers	In human primary astrocytes, *TUG1* silencing downregulated the p16 and p21 senescence markers	[50]
*EPB41L4A-AS1*	Brain aging and Alzheimer’s disease	SH-SY5Y neuroblastoma cells, U251 glioblastoma cells	None	*EPB41L4A-AS1* acetylates histones and activates gene expression required for the synthesis of NAD^+^ and ATP, which are essential for brain function	*EPB41L4A-AS1* downregulation decreased the expression levels of its target genes, also decreasing the cellular levels of NAD^+^ and ATP	[51]
*Gas5*	Alzheimer’s disease	HT22 cells, young and aged C57BL/6J mice	None	In HT22 cells, *Gas5* decreases Tau phosphorylation and increases the levels of insulin receptor and insulin signaling, which are essential for brain function	In HT22 cells, *Gas5* downregulation decreased insulin receptor levels and increased Tau phosphorylation. In HT22 cells and in the mouse hippocampus, NPC86 (a *Gas5*-binding small molecule) increased *Gas5* and insulin receptor levels	[52]
*SNCA-AS1*	Parkinson’s disease	SH-SY5Y human neuroblastoma cells	ITPR1, ITPR3, CACNA1D, FOXO1, NFATC4	*SNCA-AS1* transcription increases the mRNA and protein levels of α-synuclein	*SNCA-AS1* upregulation increased the levels of most senescence markers tested; its downregulation had opposite effects	[53]
*LincRNA-p21*	Parkinson’s disease	SH-SY5Y cells	P16, p53, telomerase activity, telomere length	MPP^+^ treatment upregulates *lincRNA-p21*, which downregulates the Wnt/β-catenin signaling pathway	*LincRNA-p21* downregulation alleviated MPP^+^-induced senescence, increased β-catenin levels, and restored senescence marker levels	[54]

Abbreviations: *Neat1*, nuclear-enriched abundant transcript 1; *Gas5*, growth-arrest-specific 5; *TUG1*, taurine-upregulated gene 1; *EPB41L4A-AS1*, erythrocyte membrane protein band 4.1 like 4A antisense 1; NAD^+^, oxidized nicotinamide adenine dinucleotide; ATP, adenosine triphosphate; *SNCA-AS1*, synuclein alpha antisense 1; ITPR1, inositol 1,4,5-trisphosphate receptor type 1; ITPR3, inositol 1,4,5-trisphosphate receptor type 3; CACNA1D, calcium voltage-gated channel subunit alpha 1 D; FOXO1, forkhead box O1; NFATC4, nuclear factor of activated T cells 4; MPP^+^, 1-methyl-4-phenylpyridinium.

**Table 2 cells-13-00119-t002:** Studies on the impact of lncRNA up/downregulation on cellular senescence in disease models of osteoarthritis and osteoporosis.

LncRNA	Associated Disease	Model Systems	Senescence Markers Assessed	Interacting Molecules and Molecular Mechanism	Up/Downregulation Outcome	Reference
*ELDR*	Osteoarthritis	Cartilage from human osteoarthritis patients, *Eldr* transgenic mice	Senescence-associated β-galactosidase, p16^INK4a^, p21, TNF-α, IL-1β, IL-6	*ELDR* binds to the IHH promoter, recruiting enzymes that methylate and acetylate chromatin	Knockdown in mice promoted chondrocyte proliferation, inhibiting senescence and cartilage degeneration	[58]
*LncZFHX2*	Osteoarthritis	Human primary chondrocytes, *lncZfhx2* conditional knockout C57BL/6 mice	Senescence-associated β-galactosidase, p16, p21	*LncZFHX2* binds to the RIF1 promoter and to the KLF4 transcription factor, recruiting KLF4 to the promoter and activating RNA transcription	* LncZFHX2 * knockdown in human chondrocytes upregulated p16 and p21. In aged mice, *lncZfhx2* overexpression attenuated cartilage degeneration. In mice, conditional *lncZfhx2* knockout in chondrocytes accelerated osteoarthritis progression	[59]
*FER1L4*	Osteoarthritis	Human blood samples, synovial fluid, and human chondrocyte cell lines	IL-6	Not investigated	Not investigated	[60]
*AC006064.4-201*	Osteoarthritis	C57BL/6 mice, human and mouse cartilage tissues	Senescence-associated β-galactosidase, p16^INK4a^, p21, and p53	*AC006064.4-201* interacts with PTBP1, blocking its binding to the CDKN1B mRNA and reducing its stability	*AC006064.4-201* knockdown in human chondrocytes upregulated senescence markers, which were reduced by *AC006064.4-201* overexpression	[61]
*Zfas1*	Osteoporosis	Bone-marrow-derived mesenchymal stem cells from young and old C57BL/6J mice	Senescence-associated β-galactosidase, p16, p53	*Zfas1* downregulates the miR-499 microRNA, further upregulating the EphA5 receptor	*Zfas1* knockdown in bone-marrow-derived mesenchymal stem cells suppressed senescence and promoted their osteogenic differentiation. In ovariectomized mice, *Zfas1* knockdown increased the bone mass	[62]
*Pmif*	Osteoporosis	Primary osteoprogenitor cells from young and aged C57BL/6J mice, MC3T3-E1 clone 14 osteoblast progenitor cell line	None	*Pmif* binds to human antigen R, inhibiting its interaction with β-actin mRNA and preventing β-actin expression, which blocks the migration of aged osteoprogenitor cells	*Pmif* knockdown promoted bone formation in aged mice	[63]
*NEAT1*	Osteoporosis	Human bone-marrow-derived mesenchymal stem cells from young and aged donors, C57BL/6J mice	P16, p21, p53	* NEAT1 * downregulates miR-27b-3p, further upregulating proteins affecting osteogenic differentiation; it also promotes CSF1 secretion, which promotes osteoclastic differentiation	Knockdown in aged mice increased bone mass	[64]

Abbreviations: *ELDR*, epidermal growth factor receptor long noncoding downstream RNA; IHH, Indian hedgehog signaling molecule; *lncZFHX2*, zinc finger homeobox 2; RIF1, replication timing regulatory factor 1; KLF4, anti-Kruppel-like factor 4; *FER1L4*, Fer-1-like protein 4; PTBP1, polypyrimidine tract-binding protein 1; CDKN1B, cyclin-dependent kinase inhibitor 1B; *Zfas1*, zinc finger antisense 1; EphA5, ephrin type-A receptor 5; *Pmif*, postulated migration inhibitory factor; *NEAT1*, nuclear-enriched abundant transcript 1; CSF1, colony-stimulating factor 1.

**Table 3 cells-13-00119-t003:** Studies on the impact of lncRNA up/downregulation on cellular senescence in model systems of cardiovascular diseases.

LncRNA	Associated Disease(s)	Model Systems	Senescence Markers Assessed	Interacting Molecules and Molecular Mechanism	Up/Downregulation Outcome	Reference
*LncR-SMAL*	Heart aging diseases	Human blood samples, C57BL/6 female mice, AC16 human cardiomyocyte cell line, HUVECs, neonatal mouse ventricular cells, neonatal mouse fibroblasts	Senescence-associated β-galactosidase, telomerase activity, p21, p53	In aged mice, *lnc-Smal* binds to Parkin, promoting its proteasome degradation and blocking the beneficial effect of Parkin on heart function	Overexpression in cardiomyocytes induced senescence, while its knockdown partially inhibited this effect	[70]
*H19*	Senescence-associated cardiac diseases	Neonatal mouse ventricular cells, aged mouse cardiomyocytes	Senescence-associated β-galactosidase, p21, p53	*H19* interacts with the miR-19a microRNA, which downregulates SOCS1. SOCS1 activates the p21/p53 signaling pathway and induces senescence	*H19* knockdown decreased senescence levels in neonatal mouse ventricular cells; overexpression had the opposite effect	[71]
*Lnc-CYP7A1-1*	Myocardial infarction	Human bone-marrow-derived mesenchymal stem cells, infarcted mice	Senescence-associated β-galactosidase, p16I^NK4a^, p27^Kip^	*Lnc-CYP7A1-1* inhibits the expression of SYNE1 and contributes to senescence	*Lnc-CYP7A1-1* downregulation in aged human bone-marrow-derived mesenchymal stem cells inhibited senescence and improved regenerative capacity after implantation in infarcted mouse hearts	[72]
*PVT1*	Cardiovascular disease	HUVECs, human serum	Senescence-associated β-galactosidase, p62, phosphorylated retinoblastoma	*PVT1* downregulates miR-31, a microRNA that binds to a promoter and activates sirtuin 3 expression, which inhibits senescence	17β-estradiol inhibited H_2_O_2_-induced senescence, an effect that was partially abolished by *PVT1* knockdown; *PVT1* overexpression inhibited endothelial senescence	[73]
*ENSMUST00000218874*	Aging-related cardiovascular diseases	C57BL/6 male mice, CRL-2797 mouse aortic smooth muscle cells	P16, p21, p53, senescence-associated β-galactosidase	*ENSMUST00000218874* potentially binds to the gene coding for adenylate cyclase 8, contributing to decreasing senescence	*ENSMUST00000218874* knockdown in mouse smooth muscle cells increased the mRNA levels of p16, p21, and p53 and decreased the mRNA levels of adenylate cyclase 8	[74]
*PCAT14*	Endothelial aging	Human endothelial cells from mesenteric arteries, HUVECs, THP-1 monocytic cells	None	Not investigated	*PCAT14* downregulation increased the expression of pro-inflammatory genes and decreased endothelial cell migratory capacity	[65]
*GAS5*	Aging-related cardiovascular diseases	Vascular smooth muscle cells of the human aorta, human serum, young and old male Sprague–Dawley rats	Senescence-associated β-galactosidase, NAD^+^/NADH ratio, γH2AX	High levels of *GAS5* downregulate the miR-665 microRNA. Decreased miR-665 levels correlate with increased levels of syndecan 1 and low levels of senescence	*GAS5* overexpression inhibited miR-665 and increased the levels of syndecan 1, which inhibited senescence in vascular smooth muscle cells	[75]
*SNHG1*	Diabetic vascular calcification/aging	Human aortic vascular smooth muscle cells	Senescence-associated β-galactosidase, p16, p21	*SNHG1* stabilizes the mRNA of the Bhlhe40 transcription factor, increasing its SUMOylation and nuclear import of the corresponding protein. In the nucleus, Bhlhe40 inhibits gene expression that otherwise results in calcification and senescence	Overexpression of *SNHG1* in high-glucose-induced human aortic vascular smooth muscle cells decreased their senescence levels	[76]
*SIRT1-AS*	Aging-related deep venous thrombosis	Blood samples from thrombosis patients, HUVECs, SAMP-1 mice	Senescence-associated β-galactosidase, p16, p21, p53	The *SIRT1-AS* antisense lncRNA upregulates *SIRT1* mRNA, which promotes FOXO3a degradation, inhibits endothelial senescence and prevents deep venous thrombosis	In HUVECs, *SIRT1-AS* overexpression upregulated *SIRT1*, alleviating deep venous thrombosis	[77]
*RAMP2-AS1*	Endothelial cell aging	HUVECS, intima samples from human mesenteric arteries	Senescence-associated β-galactosidase	*RAMP2-AS1* promotes *RAMP2* expression	*RAMP2-AS1* knockdown promoted endothelial cell senescence	[78]
*MEG8*	Cardiovascular disease	HUVECs, iPSCs	Senescence-associated β-galactosidase, p16, p21	*MEG8* binds to the CIRBP RNA-binding protein and to the HADHB protein complex, which contributes to the processing of miRNA-370 and miRNA-494	Downregulation of *MEG8* decreased the levels of miRNA-370 and miRNA-494 and promoted senescence	[79]
*OIP5-AS1*	Vascular aging	HUVECs, mice	Senescence-associated β-galactosidase, PCNA	Senescence upregulates *OIP5-AS1*, which downregulates the miR-4500 microRNA. miR-4500 binds to the 3′ untranslated region of the *ARG2* mRNA to inhibit its translation. Increased levels of ARG2 cause endothelial senescence	Depletion of *OIP5-AS1* protected against endothelial cell senescence and dysfunction	[80]

Abbreviations: *lncR-SMAL*, senescence-mitophagy-associated lncRNA; HUVECS, human umbilical vein endothelial cells; SOCS1, suppressor of cytokine signaling 1; *lnc-CYP7A1-1*, cytochrome P450 family 7 subfamily A member 1; SYNE1, spectrin repeat-containing nuclear envelope protein 1; *PVT1*, plasmacytoma variant translocation 1; *PCAT14*, prostate-cancer-associated transcript 14; *GAS5*, growth-arrest-specific 5; NAD^+^/NADH, oxidized/reduced nicotinamide adenine dinucleotide; *SNHG1*, small nucleolar RNA host gene 1; Bhlhe40, basic helix-loop-helix family member e40; *SIRT1-AS*, sirtuin 1 antisense; FOXO3a, forkhead box O3a; *RAMP2-AS1*, antisense transcript 1 of receptor-activity-modifying protein 2; *MEG8*, maternally expressed gene 8; iPSCs, induced pluripotent stem cells; CIRBP, cold-inducible RNA-binding protein; HADHB, hydroxyacyl-CoA dehydrogenase trifunctional multi-enzyme complex subunit β; *OIP5-AS1*, Opa-interacting protein 5 antisense RNA 1; PCNA, proliferating cell nuclear antigen; ARG2, arginase 2.

**Table 4 cells-13-00119-t004:** Studies demonstrating the impact of lncRNA regulation on skin aging and wound healing.

LncRNA	Associated Disease	Model Systems	Senescence Markers Assessed	Interacting Molecules and Molecular Mechanism	Up/Downregulation Outcome	Reference
*H19*	Not specified	Human dermal fibroblasts	Senescence-associated β-galactosidase, p21, p16	*H19* binds to miR-296-5p and upregulates IGF2, which activates the PI3K/mTOR signaling pathway and increases the levels of AQP3	*H19* upregulation in human dermal fibroblasts increased their viability and prevented their senescence	[84]
*MAGI2-AS3*	Not specified	Human fibroblasts	Senescence-associated β-galactosidase, p16, p21	*MAGI2-AS3* binds to the HSPA8 molecular chaperone, promoting its degradation and consequent accumulation of reactive oxygen species. Reactive oxygen species activate the MAP2K6/p38 signaling pathway, causing senescence	*MAGI2-AS3* downregulation delayed fibroblast senescence by preventing the accumulation of reactive oxygen species	[85]
*SAN*	Not specified	Human adipose-derived stem cells, fibroblasts, HUVECs, male Sprague–Dawley rats	Senescence-associated β-galactosidase, p21	*SAN* downregulates miR-143-3p, which further inhibits the expression of γ-adducin	*SAN* downregulation in adipose-derived stem cells inhibited their senescence and accelerated wound closure in transplanted rats	[86]
*LYPLAL1-AS1*	Not specified	Human adipose-derived mesenchymal stem cells, human peripheral blood	Senescence-associated β-galactosidase, p16, p21, p53, lamin B1 (negative marker)	*LYPLAL1-AS1* suppresses transcription of the miR-let-7b microRNA through binding to its promoter	*LYPLAL1-AS1* upregulation alleviated senescence	[87]

Abbreviations: IGF2, insulin-like growth factor 2; AQP3, aquaglyceroporin 3; *MAGI2-AS3*, membrane-associated guanylate kinase WW and PDZ domain-containing 2 antisense RNA 3; HSPA8, heat-shock protein family A member 8; *SAN*, senescence-associated noncoding RNA; HUVECs, human umbilical vein endothelial cells; *LYPLAL1-AS1*, lysophospholipase-like 1 antisense RNA 1.

**Table 5 cells-13-00119-t005:** Studies demonstrating the impact of lncRNA regulation on liver aging, obesity, and NAFLD.

LncRNA	Associated Disease(s)	Model Systems	Senescence Markers Assessed	Interacting Molecules and Molecular Mechanism	Up/Downregulation Outcome	Reference
*Altre*	Aging-related liver diseases	*Altre* conditional knockout mice; mouse lymphocytes from the thymus, spleen, liver, visceral adipose tissue, lymph nodes, and colon	α-Smooth muscle actin	*Altre* interacts with Yin Yang 1, regulating its binding to chromatin, further affecting the expression of genes involved in mitochondrial function and T_reg_ fitness	*Altre* downregulation in T_reg_ cells caused liver dysfunction, inflammation, fibrosis, and cancer in aged mice	[89]
*Meg3*	Obesity, NAFLD	C57BL/6J mice, human liver biopsies, HUVECs	Senescence-associated β-galactosidase, p21, p53	Obesity upregulates *Meg3* in hepatic endothelial cells. *Meg3* induces senescence in the hepatic endothelium, compromising insulin signaling and glucose homeostasis	Hepatic endothelial cell senescence induced by *Meg3* knockdown could be inhibited by hepatic endothelial-cell-specific p53 knockout	[90]
*MAYA*	NAFLD	Lakeview Golden Syrian hamsters, LO2 hepatocyte cells	Senescence-associated β-galactosidase, p16, p21, γH2AX, telomerase reverse transcriptase, telomeric repeat-binding factors 1 and 2	*MAYA* downregulates Yes-associated protein, resulting in iron overload and hepatocyte senescence, which aggravates NAFLD	Suppression of *Maya* in a NALFD mouse model alleviated high-fat-diet-induced hepatic steatosis and senescence	[91]

Abbreviations: *Altre*, T_reg_-expressed non-protein-coding RNA; *Meg3*, maternally expressed gene 3; HUVECs, human umbilical vein endothelial cells; *MAYA*, MIST-1/2 antagonizing for YAP activation; NAFLD, nonalcoholic fatty liver disease.

## Data Availability

No new data were created or analyzed in this study.

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
