# Peer review of "The Impact of Long Noncoding RNAs in Tissue Regeneration and Senescence"

_cells, 2024, doi:10.3390/cells13020119_

Round 1

Reviewer 1 Report

Comments and Suggestions for Authors

In this review, the authors propose a quite exhaustive description of the different studies showing an interest of using specific lncRNA to regulate cellular senescence for potential therapeutic impact on age-related diseases such as neurodegeneration, osteoarthritis and osteoporosis, cardiovascular diseases and liver dysfunction. A last chapter is dedicated to the technical challenges that have to be overcome for this therapeutic strategy to be efficiently used for human health. This is an overall timely written review of very good quality.

This review clearly highlights the promising potential of lncRNAs as therapeutic targets for age related diseases. I believe that some aspects of lncRNA biology should be introduced in more details. In particular, it would be interesting to have a better description of what are lncRNAs, their structure, their conservation from mouse to human which is an essential issue for the use of these elements as therapeutic tools. It would also be interesting to have a short overview on how these lncRNAs are regulated as their tissue specific expression is of paramount importance for the success of this therapeutic strategy. Similarly, some details about how lncRNAs regulate their targets should be provided to better assess the interest of these molecules as therapeutic tools.

A figure describing senescent cells and their potential impact on age-related diseases would definitely be useful to better understand their high potential as therapeutic targets.

Minor point :

 The tables are difficult to read without the lines between the description of the different lncRNAs

Comments on the Quality of English Language

Some sentences are difficult to understand:

-        Line 72 “ however … process”

-        Line 103 “this study … hippocampal function”

-        Line 152 “the Fer-1-like … disease progression”

-        Line 201 “LncRNA-SMAL … effect”

Author Response

Reviewer 1

In this review, the authors propose a quite exhaustive description of the different studies showing an interest of using specific lncRNA to regulate cellular senescence for potential therapeutic impact on age-related diseases such as neurodegeneration, osteoarthritis and osteoporosis, cardiovascular diseases and liver dysfunction. A last chapter is dedicated to the technical challenges that have to be overcome for this therapeutic strategy to be efficiently used for human health. This is an overall timely written review of very good quality.

This review clearly highlights the promising potential of lncRNAs as therapeutic targets for age related diseases. I believe that some aspects of lncRNA biology should be introduced in more details. In particular, it would be interesting to have a better description of what are lncRNAs, their structure, their conservation from mouse to human which is an essential issue for the use of these elements as therapeutic tools. It would also be interesting to have a short overview on how these lncRNAs are regulated as their tissue specific expression is of paramount importance for the success of this therapeutic strategy. Similarly, some details about how lncRNAs regulate their targets should be provided to better assess the interest of these molecules as therapeutic tools.

Response: We thank the Reviewer for this remark. To provide a robust lncRNA background, we added a new section entitled “General features of lncRNAs”(pages 2-3). Here, we describe general properties of lncRNAs, including their classification, functions, the regulation of their expression levels, and the effects they exert on their targets. Their conservation between mouse and human species is also discussed. The numbering of the subsequent sections was shifted by n+1.

A figure describing senescent cells and their potential impact on age-related diseases would definitely be useful to better understand their high potential as therapeutic targets.

Response: We thank the Reviewer for this pertinent suggestion. We added a new figure (Figure 1; previous Figure 1 has been renamed Figure 2), representing the impact of senescence-associated lncRNAs on multiple aspects of aging-related diseases, in which lncRNAs are highlighted as potential therapeutic targets. The new Figure 1 is mentioned at the beginning of the lncRNA transduction section (pages 13-14).

Minor point:

The tables are difficult to read without the lines between the descriptions of the different lncRNAs

Response: We thank the Reviewer for this observation. To improve the readability of the tables, we added horizontal and vertical gridlines to all tables.

Comments on the Quality of English Language

Some sentences are difficult to understand:

- Line 72 “ however … process”

- Line 103 “this study … hippocampal function”

- Line 152 “the Fer-1-like … disease progression”

- Line 201 “LncRNA-SMAL … effect”

Response: We thank the Reviewer for this observation on text clarity. Accordingly, we have modified these sentences, as described below:

- The sentence “However, and although lncRNAs have been shown to impact in the reprograming process, their use has important limitations, including the complexity and time requirements of cell reprogramming, cell heterogeneity among pluripotent stem cell lines and the risks of tumor formation and immune rejection after transplantation.” has been changed to “However, although lncRNAs have been shown to impact in the reprograming process, their use has important limitations. These limitations include the complexity and time requirements of cell reprogramming and heterogeneity among pluripotent stem cell lines. Moreover, the risks of tumor formation and immune rejection after transplantation should not be underestimated.”.

- The sentence “This study demonstrates the possibility of lncRNA down-regulation through physical and cognitive activities in rats, which had a positive impact on their hippocampal function.” has been changed to “This study demonstrates the possibility of lncRNA down-regulation through physical and cognitive activities in rats. These activities also had a positive impact on their hippocampal function.”.

- The sentence “The Fer-1-like protein 4 (FER1L4) lncRNA was also down-regulated in osteoarthritis patients, being its levels also decreased with aging and disease progression.” has been changed to “The Fer-1-like protein 4 (FER1L4) lncRNA was also down-regulated in osteoarthritis patients. Its levels were also decreased with aging and disease progression.”.

- The sentence “LncRNA-SMAL promoted senescence in cardiomyocytes, having its inhibition a cardioprotective effect.” has been changed to “LncRNA-SMAL promoted senescence in cardiomyocytes and its inhibition had a cardioprotective effect.”.

Reviewer 2 Report

Comments and Suggestions for Authors

The authors of the manuscript numbered by Cells-2746705 presented a review on the impact of long noncoding RNAs in tissue regeneration and senescence. By reviewing recent studies of how cellular senescence was inhibited through the up/down-regulation of specific lncRNAs, the authors thus proposed that lncRNA regulation through RNA or gene therapies may be taken as a prospective preventive and therapeutic approach against aging and multiple aging-related diseases. It seems to me that the authors only listed experimental facts or results but did not provide a deep connection among potential factors related to aging and aging-related diseases. I suggest that apart from reviewing results obtained by experimental studies, the authors should also provide a possible scheme or framework (or draw schematic diagrams for long noncoding RNAs regulations) for the modeling and analysis of how long noncoding RNAs regulate tissue regeneration or senescence. If this is possible, I think that it is helpful for theoretically analyzing and revealing the mechanism behind the impact of long noncoding RNA regulations on regeneration and senescence.

The readability of the contents in four tables is poor, e.g., the descriptions in the third, fifth and sixth columns are not separated, easily leading to misunderstandings. I suggest that “Conclusion” is revised as “Conclusion and discussion”. The “discussion” may involve modeling and analysis, besides possible connections among molecular factors related to regeneration and senescence.

Author Response

Reviewer 2

The authors of the manuscript numbered by Cells-2746705 presented a review on the impact of long noncoding RNAs in tissue regeneration and senescence. By reviewing recent studies of how cellular senescence was inhibited through the up/down-regulation of specific lncRNAs, the authors thus proposed that lncRNA regulation through RNA or gene therapies may be taken as a prospective preventive and therapeutic approach against aging and multiple aging-related diseases. It seems to me that the authors only listed experimental facts or results but did not provide a deep connection among potential factors related to aging and aging-related diseases. I suggest that apart from reviewing results obtained by experimental studies, the authors should also provide a possible scheme or framework (or draw schematic diagrams for long noncoding RNAs regulations) for the modeling and analysis of how long noncoding RNAs regulate tissue regeneration or senescence. If this is possible, I think that it is helpful for theoretically analyzing and revealing the mechanism behind the impact of long noncoding RNA regulations on regeneration and senescence.

Response: We thank the Reviewer for this interesting observation. We added a new figure (Figure 1; previous Figure 1 has been renamed Figure 2), representing the potential of lncRNAs to overcome cellular processes related to senescence and aging. In this framework, lncRNAs are potential tools to promote tissue rejuvenation and protection against aging-related diseases. The new Figure 1 is mentioned at the beginning of the lncRNA transduction section (pages 13-14).

The readability of the contents in four tables is poor, e.g., the descriptions in the third, fifth and sixth columns are not separated, easily leading to misunderstandings. I suggest that “Conclusion” is revised as “Conclusion and discussion”. The “discussion” may involve modeling and analysis, besides possible connections among molecular factors related to regeneration and senescence.

Response: We thank the Reviewer for this observation. To improve the readability of the tables, we added horizontal and vertical gridlines to all tables. We have also revised the “Conclusion” section as “Conclusion and discussion” (page 15). Here, we added that the inhibition of senescence could also be correlated with alterations in other aging hallmarks and could promote tissue regeneration, with a positive impact on aging-related diseases.

Reviewer 3 Report

Comments and Suggestions for Authors

In this article, the authors, based on a wide range of publications from recent years, review the current knowledge on the effects of various types of long noncoding RNAs (lncRNAs) on cellular senescence and tissue degeneration/regeneration, and point out the role of lncRNAs in aging-related diseases, as well as the possibility of using them in therapies for these diseases.

I found the manuscript to be well, clearly written and interesting. The information presented in it is of particular value due to its potential future use in the treatment of diseases related to aging. However, the novelty of the information included is partially limited because several review papers with very similar topics have already been published and were not included/referenced in the assessed manuscript. I have included some examples below:

He J, Tu C, Liu Y. Role of lncRNAs in aging and age-related diseases. Aging Med (Milton). 2018 Jul 30;1(2):158-175. doi: 10.1002/agm2.12030.

Kour S, Rath PC. Long noncoding RNAs in aging and age-related diseases. Ageing Res Rev. 2016 Mar;26:1-21. doi: 10.1016/j.arr.2015.12.001.

Kim C, Kang D, Lee EK, Lee JS. Long Noncoding RNAs and RNA-Binding Proteins in Oxidative Stress, Cellular Senescence, and Age-Related Diseases. Oxid Med Cell Longev. 2017;2017:2062384. doi: 10.1155/2017/2062384.

Ni YQ, Xu H, Liu YS. Roles of Long Non-coding RNAs in the Development of Aging-Related Neurodegenerative Diseases. Front Mol Neurosci. 2022 Mar 14;15:844193. doi: 10.3389/fnmol.2022.844193

Author Response

Reviewer 3

In this article, the authors, based on a wide range of publications from recent years, review the current knowledge on the effects of various types of long noncoding RNAs (lncRNAs) on cellular senescence and tissue degeneration/regeneration, and point out the role of lncRNAs in aging-related diseases, as well as the possibility of using them in therapies for these diseases.

I found the manuscript to be well, clearly written and interesting. The information presented in it is of particular value due to its potential future use in the treatment of diseases related to aging. However, the novelty of the information included is partially limited because several review papers with very similar topics have already been published and were not included/referenced in the assessed manuscript. I have included some examples below:

He J, Tu C, Liu Y. Role of lncRNAs in aging and age-related diseases. Aging Med (Milton). 2018 Jul 30;1(2):158-175. doi: 10.1002/agm2.12030.

Kour S, Rath PC. Long noncoding RNAs in aging and age-related diseases. Ageing Res Rev. 2016 Mar;26:1-21. doi: 10.1016/j.arr.2015.12.001.

Kim C, Kang D, Lee EK, Lee JS. Long Noncoding RNAs and RNA-Binding Proteins in Oxidative Stress, Cellular Senescence, and Age-Related Diseases. Oxid Med Cell Longev. 2017;2017:2062384. doi: 10.1155/2017/2062384.

Ni YQ, Xu H, Liu YS. Roles of Long Non-coding RNAs in the Development of Aging-Related Neurodegenerative Diseases. Front Mol Neurosci. 2022 Mar 14;15:844193. doi: 10.3389/fnmol.2022.844193

Response: We thank the Reviewer for this important remark. In the introduction, we have cited the four articles mentioned above (references #26 to 29). Furthermore, we would like to stress that the main focus of our work is the up/down-regulation of senescence-associated lncRNAs. This prospective therapeutic approach has not been explicitly discussed in any of the four articles mentioned above. As such, we are confident that our manuscript stands out from other reviews exclusively focused on the pathophysiological impact of lncRNAs in senescence and aging-related diseases.

Round 2

Reviewer 2 Report

Comments and Suggestions for Authors

The authors have responded to my comments. I has no additional comments, but suggest that the authors check their manuscript again, focusing on English writing and spelling.